# Exploring Nurse Dyads' experiences of scope of practice in nursing homes: A qualitative descriptive study from the FLORENCE project

Ole Martin Nordaunet[1,2]*, Hanne Aagaard[1], Cecilia Olsson[1,2], Edith Roth Gjevjon[1,3], Gunilla Borglin[1]

1 Department of Bachelor Education (Nursing), Lovisenberg Diaconal University College, Oslo, Norway, 2 Department of Nursing, Institute of Health Sciences Karlstad University, Karlstad, Sweden, 3 Department of Bachelor Education, The Arctic University of Norway, Harstad, Norway

* Olemartin.nordaunet@kau.se

## Abstract

### Background

A nurse's scope of practice includes the full range of roles, functions, responsibilities, and decision-making authority. However, how this scope—particularly in relation to the fundamentals of care—is experienced by registered nurses and non-registered nurses in nursing homes remains underexplored. This study aimed to explore the experiences of nurses regarding their mutual general scope of practice, the differences in their individual scope of practice, and their respective scopes in relation to the fundamentals of care in the context of nursing homes.

### Methods

This qualitative study included eight nursing dyads (n = 16) from four nursing homes in south-east Norway. Data were collected through individual, focused interviews, i.e., non-dyadic data, and analysed using the framework method for dyadic analysis. The study followed the Standards for Reporting Qualitative Research (SRQR).

### Results

Registered nurses described emotional strain due to a gap between professional ideals and the realities of high-pressure, understaffed environments. The contrast between being perceived as "too posh to wash" and "too busy" highlighted how systemic factors, rather than personal attitudes, shaped role perceptions. These constraints influenced care delivery and undermined their professional identity. Registered nurses tended to focus on indirect care and physical needs, while non-registered nurses took on more direct care, particularly the fundamentals of care—physical, relational, and psychosocial. Although the registered nurses valued

**Data availability statement:** As outlined in the consent forms provided to participants, all informants in this study are entitled to confidentiality and anonymity. Consequently, the institutional Data Protection Officer has recommended that the data be shared only upon request. As the data contain potentially identifying and sensitive information, access will only be granted for justifiable reasons and will be subject to a signed confidentiality agreement and anonymisation procedures for the requested data. This approach complies with the Personal Data Act of Norway (Ministry of Justice and Public Security, 2018), the National Strategy on Access to and Sharing of Research Data (Ministry of Education and Research, 2018), and GDPR guidelines. Requests for data access can be directed to the institutional Data Protection Officer at personvernombud@ldh.no, who will evaluate inquiries on a case-by-case basis.

**Funding:** The author(s) received no specific funding for this work.

**Competing interests:** The authors have declared that no competing interests exist.

**Abbreviations:** RN: Registered Nurses; Non-RN: Non-registered Nurses; SoP: Scope of Practice; FoC: Fundamentals of Care.

holistic care, frequent delegation to the non-registered nurses raised concerns about care quality and role clarity.

## Conclusion

Registered nurses' scope of practice was shaped more by workload demands than by reluctance to engage in the fundamentals of care. The dyadic approach provided new insights into how professional responsibilities and activities are co-constructed in complex care settings. Findings highlight the need for organisational and clinical strategies to clarify role boundaries, strengthen registered nurses' professional identity, reinforce the delivery of the fundamentals of care, and support effective and safe delegation practices.

## Introduction

In recent years, the concept of nurses' scope of practice (SoP) has gained prominence on the research agenda as a means to better delineate and define nursing roles. In this study, the term *"nurses"* is used to refer to both registered nurses (RNs) and non-registered nurses (non-RNs). More recently, SoP has been defined as "the full spectrum of roles, functions, responsibilities, activities, and decision-making capacity which nurses are educated, competent, and authorised to perform" [1] (p. 7). This definition underscores that the SoP should reflect the full breadth of nursing capabilities, with particular emphasis on the role, function, and decision-making unique to each type of nurse and their qualifications [2]. SoP can thus be understood as grounded in the individual nurse's professional stance, reflective capacity, and practical problem-solving skills [3], forming the theoretical and practical foundation for professional nursing.

Despite increasing calls to optimize the use of nursing resources, research continues to show that nurses' SoP remains underutilized or poorly aligned with their full potential [4,5]. This can be viewed within the broader context of systemic changes in healthcare delivery—particularly the shift wherein bachelor-educated RNs have become less visible at the bedside, while less formally trained personnel, such as non-RNs, are increasingly engaged in tasks traditionally performed by RNs. This shift has reoriented the RN's role toward supervision, coordination, and administrative oversight [6]. Such changes are particularly relevant for what is arguably the most frequently provided aspect of nursing care: the fundamentals of care (FoC) [7,8].

SoP in relation to the FoC encompasses nurses' responsibility to address the complex and interrelated physical, psychosocial, and relational needs of patients. This framing places the nurse's commitment to care at the centre, situating SoP within the broader care context [9]. A clear understanding of SoP is especially important given the involvement of multiple nursing roles—including RNs and non-RNs—in delivering fundamental care. Without this clarity, role confusion may occur, professional identity may be undermined, and there is an increased risk of nurses working outside their authorised scope [5,10,11]. This complexity is further exacerbated by the ongoing

global shortage of RNs [12], prompting the development of new care models that involve task-shifting [13]. Such models often expand non-RNs' responsibilities while assigning RNs broader oversight over patient safety, coordination, and quality assurance [2,14,15]. The implications of these shifts are particularly significant in facility-based care settings, where skill mix is a recognised patient safety factor [16]. Additional factors such as nurse-to-patient ratios and time spent per resident [17] further modulate care outcomes. Although Norway has a well-developed facility-based care infrastructure, characterised by a high presence of non-RNs at the bedside [18], little research has examined how this affects patient safety and care quality—especially in relation to FoC, despite its clinical significance [19,20].

Given that RNs and non-RNs are the primary providers of care, these two professions collaborate closely, each contributing within their respective SoP. While the professional relationship between them is outlined in regulatory guidelines [6,21], non-RNs have often been described as the "operational arm" of RNs [22]. In this dynamic, the RN's role encompasses leadership, coordination, assessment, care planning, decision-making, delegation, and supervision [1,23,24], making them accountable for the quality of care provided. However, the RN's SoP extends beyond administrative responsibilities to include the direct delivery of FoC—historically central to nursing and still vital for ensuring safe and optimal care [25,26]. A preliminary literature review revealed a notable gap: while the SoP of specialist RNs is well studied, less is known about the everyday roles and activities of generalist RNs—especially in nursing homes. Few studies examine how SoP is enacted in relation to the FoC in facility-based care [19,27], and even fewer explore how RNs and non-RNs perceive and navigate their respective SoPs to address FoC.

## Aim

This study aimed to explore the experiences of nurses regarding their mutual general scope of practice, the differences in their individual scope of practice, and their respective scopes in relation to the fundamentals of care in the context of nursing homes.

## Methods

A qualitative descriptive design [28] was chosen as a pragmatic approach to gather rich, contextualised data. Data were collected through individual focused interviews [29], and analysed using the framework method, inspired by Collaço et al. [30]. This method enabled exploration of the variations in nurses' experiences based on their professional responsibilities and activities—i.e., their SoP—while also identifying shared concerns, similarities, and incongruences both within and across professions. The study is reported in line with the Standards for Reporting Qualitative Research (SRQR) [31].

### Recruitment and informants

Informants were nurses employed at four nursing homes in southeast Norway. In Norwegian facility-based care, three types of nursing personnel are present: RNs, health care workers, and health care assistants. RNs hold a bachelor's degree in nursing, which provides licensure, and the skills needed to lead care for all patients, including those with complex conditions [32]. Health care workers, considered non-RNs here, have completed upper secondary education plus apprenticeship and licensure [33]. Health care assistants, lacks formal health education, and make up about 25% of the nursing workforce [34].

The informants had previously consented to take part in a larger two-part research project. The first part involved structured observations [35], while the current study used a qualitative descriptive design with focused interviews. From 58 eligible nurses (24 RNs and 34 non-RNs), 21 RNs and 10 non-RNs were purposefully selected and invited to participate [36]. Reasons for non-participation included time constraints, language barriers, and one withdrawal due to personal reasons. From the recruited nurses, eight RNs and eight non-RNs were selected to form eight dyads. The selection was based on having worked together in the same unit and on the same shift during the structured observations in the first part of the project [36].

## Data collection

Data were collected through 16 individual, focused interviews, representing non-dyadic qualitative data [30]. Conducting individual interviews helped to mitigate potential challenges such as dominance by one participant, which can occur in dyadic settings due to differences in social status, confidence, or assertiveness [37]. Interviews were conducted at a mutually agreed time and place, audio-recorded, and demographic/background information was also collected. The interviews centred around two open-ended questions:

1. "Can you describe your nursing practice in general and in relation to FoC?"

2. "Can you describe your individual and shared scope of nursing practice?"

Probing questions such as "Can you give me an example?", "Can you explain in more detail?", and "How did you think, feel, or reflect?" were used to encourage elaboration. All interviews were transcribed verbatim using GDPR-compliant transcription software [38] and secure data storage [39]. Data collection took place between March and June 2024. The average length of the 16 interviews was 52 minutes.

## Data analysis

The non-dyadic qualitative interview transcripts were analysed using a dyadic approach informed by the Framework Method. Conducting separate individual interviews (i.e., non-dyadic data) allowed us to perform analysis at both the individual and dyadic levels—an approach not possible with interviews conducted at the same time with both informants (i.e., dyadic qualitative data) [30]. This strategy enabled us to examine contrasts and overlaps between each individual dyad's perspectives, exploring both the explicit content (text) and implicit meanings (subtext) at descriptive and interpretive levels [40].

The analysis followed eight iterative stages. Stage 1 involved transcription of the interviews, followed by familiarisation with the data (Stage 2). In Stage 3, content relevant to the interview foci was extracted and entered into a structured dyadic coding template. To assess interrater reliability, one dyad was randomly selected. Each author independently extracted quotations in response to interview questions 1 and 2 from the first three pages of the nurses' interviews. This process yielded a 93.3% agreement rate. In Stage 4, codes were organised into a chart, which informed the analytical process in Stage 5. This stage laid the foundation for developing an analytical framework in Stage 6. The framework was refined iteratively in Stage 7 and then used to guide the interpretation of the data in Stage 8, drawing on insights gathered throughout all preceding stages (Table 1).

Stages 1–4 were conducted separately for each professional group. From Stage 5 onwards, codes were compared across professions to identify patterns of similarity and difference both within and between groups. Thematic development and refinement continued iteratively as our familiarity with the data deepened. The authors collaborated closely at each stage, ensuring rigorous and consistent progression through the analytic process.

## Ethical considerations

The study received ethical approval from relevant authorities in both Norway and Sweden:

• Regional Committees for Medical and Health Research Ethics, Norway (ID: 691021/2024)

• Norwegian Agency for Shared Services in Education and Research (ID: 524471/2024)

• Research Ethics Committee, Karlstad University, Sweden (ID: C2024/55)

• Swedish Ethical Review Authority (ID: 2024-000586-01)

Informants were recruited between 28 February and 4 June 2024. All informants provided written informed consent after receiving both verbal and written information about the study.

**Table 1. Overview of the process for the framework analysis.**

| General theme - SoP in relation to residents´ general care | | | | | |
|---|---|---|---|---|---|
| **DYAD 1** | **RN** | | **Non-RN** | | |
| *Interview Question* | *Quote* | *Coding* | *Quote* | *Coding* | *Characteristics* |
| 1 | "I feel that my role is to have a bit of overview… one become very focused on the medical part… and to bring the medical aspect into the care part." | Keeping the overview Focused on medicinal care Bringing medical aspect into care | "I am here to help those who are here, those who can't manage at home themselves. Those who don't have any relatives to help them. Their daily needs. To dress them and feed them and make sure they are well…. I would say we are caregivers. We are here to help the patients with their needs, both daily, spiritual, social and physical needs." | Helping those who can´t manage Daily needs Hygiene and dressing Eating and drinking Holistic care | *The why´s in general SoP The what's and the how's in general SoP* |
| **General theme – SoP in relation to the residents´ FoC** | | | | | |
| **DYAD 4** | **RN** | | **Non-RN** | | |
| *Interview Question* | *Quote* | *Coding* | *Quote* | *Coding* | *Characteristics* |
| 1 | "I arrange conversations with the relatives and then I do it twice a year. Other tasks, for example as a nursing task... Ensuring that the residents are well here. In terms of hygiene, like medications… We create measures so they can go for a walk once, two or three times a week." | Involve and inform relatives Hygiene and dressing Medication management Mobility Create tasks in care plan | "I often say, 'good morning' and 'did you sleep well?'. Many of them recognize me, I would say. It's very nice. I also sit a bit at the bedside. Someone got a cup of coffee at the bedside. A bit of contact before we start the care… I make sure to wash well… Make sure the water is warm… Make sure they sit on the toilet… There's care and medication and... Yes, if you think like that, yes. Care and medication." | Introducing, friendly Sitting down bedside Eating and drinking Hygiene and dressing Medication management | *The what's of the FoC* |
| **General theme - Notions about each-other's SoP** | | | | | |
| **DYAD 8** | **RN** | | **Non-RN** | | |
| *Interview Question* | *Quote* | *Coding* | *Quote* | *Coding* | *Characteristics* |
| 2 | "The difference between RN and non-RN is actually nothing... The only thing is that nurses are mostly focused on medication management. Otherwise, the rest is the same. No difference, except you have more oversight over medications…" | Little difference between RN and non-RN RN have more competence in relation to medication management | "Nursing practice, yes... We also have very good cooperation with them, and it is about helping each other, of course. I think it's fine when we call if a patient is unwell, if a patient needs some medication or extra painkillers like strong medications. If we take the medications, but we can't give the strong medications, they come, or they bring the medications. That's what we have. Good cooperation." | Collaborating with RNs Call the RN is patient is unwell RNs bring medicines that non-RN can´t administer | *The "other" why´s The "other" what's and the how's Absences vs. presence* |

**Abbreviations:** FoC = Fundamentals of Care; Non-RN = non-Registered Nurse; SoP = Scope of Practice; RN = Registered Nurse.

## Rigour and reflexivity

All authors are RNs. Four hold academic positions and have extensive experience in clinical and research settings. The first author is a PhD student with no prior detailed experience in the nursing home context; the same applies to two other team members. The remaining two authors possess extensive experience in facility-based care and geriatric nursing. This diverse composition contributed to critical reflection throughout the research process. Regular discussions and reflexive practices ensured that interpretation remained grounded in the informant's narratives, minimising the impact of preconceptions.

## Results

Eight dyads participated in this study. The RNs consisted of five females and three males, with a mean age of 37.7 years (range: 28–58, SD = 11). The eight participating non-RNs were all female, with a mean age of 46.7 years (range: 32–61, SD = 10.2). The overall mean age of the informants was 42.2 years (Table 2). Both RNs and Non-RNs had about 7 years of working experience (mean) and three out of eight in both groups had undergone additional educational courses.

The analysis of the transcribed texts resulted in five general mainly congruent themes reflecting the nurses' experience of their individual and mutual general SoP, as well as their SoP related to the FoC among older people at nursing homes. Additionally, one general theme was identified, mirroring RNs' experiences of challenges related to 'the others'.

### Theme 1: The context of practice

The first mutual theme, "The context of practice," reflected variations in nurses' experiences of their general SoP, while also revealing underlying systemic challenges. This theme comprised three key characteristics: duties versus chores, dwindling resources, and environmental challenges. Each of these were interpreted as modulating the nurses' SoP.

The distinction between duties vs chores highlighted how the nurses experienced their responsibilities differently. RNs reported feeling trapped in a complex organisational system dominated by mandatory administrative obligations. These included tasks related to the nursing home's internal quality systems, as well as broader municipal and governmental care indicators. They described their responsibilities as representing a "fork in the road": on one side was their professional duty to ensure high-quality care for residents; on the other, the demands of the system consumed valuable time, creating a sense of frustration and detachment from direct patient care. These obligations were experienced as bureaucratic and exasperating. In contrast, non-RNs referred to their responsibilities as daily "chores." Although their accounts echoed concerns about time being diverted from care, they focused more on routine operational duties such as maintaining

**Table 2. Informant characteristics.**

| Dyads | Age (years) | Sex | Working experience (years) | Further education |
|---|---|---|---|---|
| Dyad I: RN | 58 | ♀ | 11 | N/A |
| Dyad I: Non-RN | 37 | ♀ | 4 | N/A |
| Dyad II: RN | 34 | ♀ | 7 | N/A |
| Dyad II: Non-RN | 32 | ♀ | 11 | Dementia and person-centred care |
| Dyad III: RN | 41 | ♂ | 6 | Administration and leadership |
| Dyad III: Non-RN | 55 | ♀ | 3 | N/A |
| Dyad IV: RN | 30 | ♂ | 6 | Mental health |
| Dyad IV: Non-RN | 61 | ♀ | 17 | Mental health |
| Dyad V: RN | 34 | ♀ | 8 | N/A |
| Dyad V: Non-RN | 57 | ♀ | 10 | N/A |
| Dyad VI: RN | 34 | ♂ | 1 | N/A |
| Dyad VI: Non-RN | 41 | ♀ | 2 | N/A |
| Dyad VII: RN | 43 | ♀ | 14 | Advanced Practice Nurse |
| Dyad VII: Non-RN | 44 | ♀ | 9 | N/A |
| Dyad VIII: RN | 28 | ♀ | 4 | N/A |
| Dyad VIII: Non-RN | 46 | ♀ | 3 | Dementia care and geriatric psychiatry |

**Abbreviations:** RN = Registered nurse; Non-RN = non-registered nurse; N/A = Not applicable.

cleanliness, supporting mealtimes, ensuring a well-functioning ward environment, and completing required documentation. This included updating quality indicators and conducting mandatory functional assessments. Despite differing roles and language, the nurses conveyed a shared perception that systemic demands significantly limited their ability to spend meaningful time with residents (S2:1).

The nurses reported that dwindling resources—particularly related to staffing levels and skill mix—negatively affected their SoP. RNs primarily expressed concern about the shortage of RNs and the challenges of working in under-staffed units. One RN described the situation starkly: "If everyone breathes at the end of the shift, you are satisfied" (Dyad 1), highlighting the high-pressure environment and reduced capacity to provide comprehensive care. Non-RNs also addressed issues related to staffing, but their concerns encompassed a broader range of factors. In addition to echoing the RNs' concerns about inadequate staffing and the impact on care quality, non-RNs described the implications of limited budgets, noting how financial constraints restricted staffing flexibility and compromised the overall safety and well-being of residents. Their accounts illustrated a wider perspective on systemic limitations and resource allocation (S2:2).

The theme *"context of practice"* also highlighted environmental challenges faced by the nurses. While the concerns were similar, non-RNs expressed a broader range of issues. RNs primarily described their SoP as hectic and stressful, marked by time pressure and a constant need to move between tasks. In contrast, non-RNs not only shared these concerns but also reported feeling the additional burden of an overwhelming workload. Their experiences reflected a sense of exhaustion, with many describing the feeling of carrying the entire workload on their own, which contributed to a heightened sense of pressure and fatigue (S2:3).

**Theme 2: The Nurses' professional stance**

The theme "the nurses' professional stance" reflected a mutual understanding in how nurses experienced their general SoP but also highlighted incongruencies in two main characteristics: Professional conduct and professional values vs. working morale, which were interpreted as constituting the theme.

The RNs described their SoP in a way that reflected professional conduct characterized by taking shortcuts, even though they knew how to perform tasks properly and were aware of the expected practices. The reasons they provided for this behaviour included a desire to avoid burnout, lack of time to do things properly, or feeling that they lacked the capacity to do so. One RN from Dyad 1 stated:

> *"So, you overlook that thing because another thing is more important, in a way. And because you can't be bothered to fight all the battles all the time. And that you're a bit lazy. I don't know."*

In contrast, the Non-RNs expressed a SoP where their professional conduct was in clear incongruence with the RNs' conduct. The non-RNs emphasized the importance of teamwork, collaborative efforts, and the need to cooperate more, regardless of professional titles. They also highlighted the importance of individual conduct, which included being ethically considerate, empathetic, contributing to a positive working environment, and taking responsibility. Furthermore, they saw it as part of their general SoP to support RNs by easing tasks such as medication management, staying professional in challenging situations, and displaying pride in their work (S2:4).

The theme 'nurses' professional stance' also encompassed experiences related to the nurses' professional values vs. their working morale. This characteristic was experienced by the nurses in ways that were mutual yet reflected some variation. RNs described a sense of never feeling that their work was good enough, often not feeling like a competent nurse, experiencing guilt, and striving to do their best to get everything done. Non-RNs described experiences similar to RNs, differing mainly in their view of responsibilities. They highlighted concerns about compromised professional values and a care culture they saw as ethically questionable (S2:5).

### Theme 3: SoP in relation to Residents' general care

The mutual theme, 'SoP in relation to residents' general care', highlighted an incongruency across two main characteristics: The whys and the what's.

The first characteristic, the whys, primarily reflected the RNs' descriptions of their professional responsibilities, particularly their role. RNs expressed that they were tasked with solving problems on a daily basis, managing the overall daily operation, and ensuring that residents' needs were met in an adequate manner. Their experience also included responsibilities for prioritizing activities and tasks, as well as transferring accountability—such as delegating tasks to others. RNs noted that they often had to forward unfinished tasks to the next shift. Non-RNs, by contrast, described assisting RNs with minor tasks and spending more direct time with residents—offering emotional support, sitting with them, and holding hands (S2:6)."

The second characteristic, the what's, predominantly reflected the activities and tasks, or their function, as described by the nurses. These were seen as part of their general SoP within the nursing home context. They described their primary engagement in clinical observations and assessments. Additionally, the non-RNs emphasized their involvement in delivering daily care, such as addressing residents' physical needs and supporting the maintenance of their functional abilities. They also described activities such as forwarding information to the RNs and updating documentation in residents' journals as part of their SoP (S2:7).

### Theme 4: SoP in relation to the Residents' FoC

The fourth theme, 'SoP in relation to the residents' FoC', reflected a sense of mutuality between the nurses, yet also revealed substantial variation and some concerns within the single characteristic that constituted this theme: The what's.

The nurses described their activities and tasks—their what's, or functional roles—in ways that predominantly focused on residents' physical needs. For the RNs, their SoP in relation to FoC was described as having a strong somatic focus, with medication management consistently prioritized. In contrast, the non-RNs' descriptions varied more substantially and demonstrated a broader engagement with the psychosocial and relational dimensions of care. While their activities in the physical domain included assisting with hygiene, dressing, nutrition, and monitoring medication, they also emphasized contributions within the psychosocial and relational domains. These included ensuring residents' well-being, offering emotional support—such as holding hands, showing compassion, providing comfort and building trust—and involving relatives in the care process (S2:8).

RNs and non-RNs shared a mutual concern regarding the what's—the activities and tasks—that reflected a significant shift in the nursing home population, with residents now being older and frailer than in previous years. This shift was particularly notable when considered alongside their earlier accounts in the theme 'The context of practice', where they described challenges such as inadequate staffing and a lack of appropriate skill mix. These systemic limitations were seen as increasingly problematic in light of the growing number of residents requiring care across all dimensions of the FoC—namely, physical, psychosocial, and relational needs.

Among all identified themes, 'SoP in relation to the residents' FoC revealed the largest incongruency between the nurses what's, i.e., the specific activities and tasks they reported engaging in. (See pattern chart inspired by Waigwa et al., 2018 [41]–Table 3-based on the FoC framework [8,9].

The pattern chart helped visualize that RNs predominantly described their SoP as aligned with the physical dimension of the FoC framework, with medication management being the most frequently reported activity. In contrast, non-RNs described a more varied SoP, encompassing activities and tasks across all three dimensions of the FoC framework—physical, psychosocial, and relational. This variation points to a divergence in care priorities and engagement between the two nursing roles within the same care context.

 

**Table 3. Pattern chart of interview codes for SoP in relation to the FoC framework [8,9].**

| FoC dimensions | Dyad I | | Dyad II | | Dyad III | | Dyad IV | | Dyad V | | Dyad VI | | Dyad VII | | Dyad VIII | |
|---|---|---|---|---|---|---|---|---|---|---|---|---|---|---|---|---|
| | RN | Non-RN | RN | Non-RN | RN | Non-RN | RN | Non-RN | RN | Non-RN | RN | Non-RN | RN | Non-RN | RN | Non-RN |
| *FoC activities related to psychical needs:* | | | | | | | | | | | | | | | | |
| - Rest and sleep | | x | | x | | x | | | | | | | | | | x |
| - Personal cleansing and dressing | | | x | | | x | | | | | | | | x | | x |
| - Medication management | x | x | | x | x | x | x | x | x | | x | x | x | x | x | x |
| - Toileting needs | | | | x | | x | | x | | | | | | x | | |
| - Eating and drinking | | x | x | | x | x | x | x | | x | x | | | x | | x |
| - Comfort including vital parameters, pain breathing and positioning | | x | | | x | x | x | x | x | | | | | x | | |
| - Safety, e.g., risk assessment, risk prevention and hygiene | | x | | x | | | | x | | | | | | x | | |
| - Mobility | | x | | x | x | | x | | | | | | | | | x |
| *FoC activities related to psychosocial needs:* | | | | | | | | | | | | | | | | |
| - Verbal and written information adjusted to the patient | x | x | | x | x | x | | x | | | x | | | x | x | x |
| - Being involved and informed about the care | | x | | x | | | | x | | | | | | | | x |
| - Respected as a person | | | | | | x | | x | | | | | x | | | |
| - Get education and information | | | | | | | | | | | | | x | | | |
| - Values and beliefs being considered and respected | | x | | x | x | x | x | x | | | | | | | | |
| - Dignity not being embarrassed or offended | | x | | | | | | | | | | | x | | | x |
| - Emotional wellbeing, anxiety, worry or stress | | | | | | | | | | | x | x | | x | | |
| - Right to privacy, not speaking above the patient | | | | | x | | | x | | x | | | | | | |
| *FoC activities related to relational needs:* | | | | | | | | | | | | | | | | |
| - Being emphatic, understanding the patient's situation | | | | | | | | | | | x | | x | | x | |
| - Helping patients to cope, supporting in developing and maintaining coping strategies | | x | | | | | | | | | x | | | | | |
| - Engage with the patient, seeing them as individuals | | | | x | x | x | x | x | | | x | x | x | x | x | x |
| - Supporting and involving family and carers | x | x | | x | x | x | | | | | | | x | x | x | x |
| - Working with the patient to set goal | | | | | | | | | | | | x | x | | | |
| - Active listening, showing responsiveness | | | | | | | | | | | | | | | | |

*(Continued)*

**Table 3.** (Continued)

| FoC dimensions | Dyad I | | Dyad II | | Dyad III | | Dyad IV | | Dyad V | | Dyad VI | | Dyad VII | | Dyad VIII | |
|---|---|---|---|---|---|---|---|---|---|---|---|---|---|---|---|---|
| | RN | Non-RN | RN | Non-RN | RN | Non-RN | RN | Non-RN | RN | Non-RN | RN | Non-RN | RN | Non-RN | RN | Non-RN |
| - Helping the patient to stay calm | | | | | | | | | | | | | | | | |
| - Being compassionate, show care for the patient | | ✗ | | | | ✗ | | ✗ | | | ✗ | ✗ | ✗ | ✗ | | |
| - Being physically and mentally present with the patient | | ✗ | | | | | | | | | ✗ | ✗ | | | | |
| *FoC activities displaying a commitment to care as in a trusting nurse-patient relationship* | | | | | | | | | | | | | | | | |
| - Developing relationship, Inviting, introducing him/herself, friendliness | | ✗ | | | | ✗ | | ✗ | | | ✗ | ✗ | ✗ | ✗ | ✗ | ✗ |
| - Focusing, eye contact, body language, interpreting signals | | ✗ | | | | | | ✗ | | | | | | ✗ | | |
| - Anticipating patients need before the emerge, reflecting prior experiences | | | | | | | | ✗ | | | | | | ✗ | | |
| - Knowledge about health history, fundamental needs and the patients' personal preferences | | ✗ | | | ✗ | | | | | | | | | | | |
| - Evaluating care in dialogue with patient | | | | | | | | | | | | | | | ✗ | ✗ |

**Abbreviations:** RN = Registered nurse; Non-RN = Non-registered nurse.

## Theme 5: Notions about each other's SoP

The theme 'Notions about each other's SoP reflected varied and incongruent descriptions, as well as concerns regarding how the nurses experienced and described each other's SoP. Three overarching characteristics constituted the theme: The 'other's' whys, the 'other's' what's, and absence vs. presence.

The RNs' descriptions of the other's whys—that is, the non-RNs' responsibilities in relation to their own—reflected concerns related to their own vague and unclear role. They described a struggle to articulate their role, using expressions such as: "it is too difficult to explain," "the role is unclear," "there's a lack of boundary," or "there is not much difference between us —only medication." Their experiences also highlighted how limited resources and economic constraints led to task shifting, which further complicated their role. When RNs did describe their responsibilities, it was mainly in terms of having more responsibility overall, such as medication management, maintaining an overview, performing administrative tasks, planning and ordinating care, and conducting more complex or specialised procedures. In contrast, the non-RNs' descriptions of the other's whys also reflected difficulty in articulating the RNs' role, with some suggesting the RNs depended on the non-RNs in practice, while others expressed that non-RNs themselves relied on RNs' clinical knowledge. There were also descriptions indicating that non-RNs sometimes had to take responsibility for activities outside of their expected role, even though the RNs were still involved in point-of-care activities (S2:9).

Descriptions of the other's what's—that is, the activities and tasks performed—also reflected inconsistencies. The RNs predominantly focused on their own SoP, especially the preparatory work surrounding physician visits. They described activities such as preparing cases, managing medication, taking bloodwork, informing about residents' status, and executing delegated tasks. Some RNs expressed frustration over being assigned to what they perceived as less relevant tasks, and a concern that non-RNs assumed they were only "sitting in front of the computer" rather than actively preparing for physician visits or revising care plans.

The non-RNs' descriptions of the RNs' activities mostly aligned with those of the RNs, including involvement in medication management and physician-related preparations. Non-RNs acknowledged that RNs had a greater workload but also noted that RNs were less visible in daily care. When non-RNs described their own activities in relation to the RNs, they emphasised monitoring, documenting, and relaying information to the RNs, as they were more consistently present with the residents (S2:10).

The final characteristic of this theme, absence vs. presence, reflected experiences of physical presence in the ward and closeness to care. RNs described frequently working alone, being less physically present on the units, and having responsibility for many residents—sometimes for an entire unit, or even the whole nursing home during weekends. This limited presence made it difficult for them to engage directly in care. The non-RNs shared similar experiences, describing the RNs as less present during shifts, and more distanced from daily resident interaction.

They also described that RNs often entered the unit only when called and were otherwise mostly absent, reinforcing the view that non-RNs were the ones consistently closest to the residents (S2:11).

## Theme 6: Challenges related to the 'Others'

The final theme, 'Challenges related to the others', was identified as a unique theme, emerging solely from the RNs' experiences. Although this theme was not reflected in the non-RNs' accounts, it was considered too significant to omit, given its implications for collaborative practice and care quality. The theme centred around one key characteristic: not stronger than our weakest link.

This characteristic described the RNs' experiences of dependence on non-RNs when accountability was transferred, such as when delegating clinical activities or care responsibilities. RNs described being acutely aware of the varying skills and competencies among non-RNs, which could affect their ability to assess situations and make informed decisions. They also emphasised the importance of trust—trusting non-RNs to monitor residents accurately, to communicate changes clearly, and to complete delegated tasks properly. The RNs highlighted that, in many cases, they were

required to rely on non-RNs for key aspects of resident care—particularly given persistent RN shortages—and that this created a vulnerability in the system. One RN expressed this dependence, and the challenges associated with trust in the following way:

> *"A healthcare worker who has taken a course so that they are allowed to administer medication, you can't go and check them afterwards, you have to trust that it is done." (Dyad 2, RN)*

The theme also encompassed RNs' descriptions of how delegation often occurred as a response to understaffing. Activities described as "minor"—such as wound care, basic clinical observations, medication management, and addressing psychosocial and relational needs, were commonly delegated. Some RNs found delegation difficult, noting that certain non-RNs resisted accepting tasks that they perceived as outside their role. These challenges were particularly evident in relation to the organisational context, and sometimes caused frustration for the RNs:

> *"That's the problem because... It is also a challenge in terms of management. Because there aren't many healthcare workers who accept tasks from us... They say that it is not their job. Many say that. 'Those are nursing tasks. It is not so positive, when we delegate tasks to them, it is challenging…" (Dyad 5, RN)*

This theme underscored a critical concern among RNs: although they held ultimate responsibility for resident care, their ability to fulfil this role depended heavily on the skills, cooperation, and willingness of the non-RNs to whom tasks were delegated. This reliance exposed a vulnerability in their scope of practice, as they were "not stronger than the weakest link."

## Discussion

Using the framework method, we identified six overarching themes: (1) the context of practice, (2) the nurses' professional stance, (3) the SoP in relation to residents' general care, (4) SoP in relation to residents' FoC, (5) notions about the other's SoP, and (6) challenges related to the other. These themes reflected the nurses' experiences of their individual and mutual general SoP, as well as their SoP related specifically to the FoC. Each theme was composed of multiple sub-characteristics that offered nuanced insights into the nurses' experiences. Below, we focus on the most salient findings.

Unsurprisingly, the professional practice environment emerged as a key factor *modulating* the nurses' *SoP*. A range of environmental modulators characterised the nurses' experiences, including excessive administrative tasks, repetitive routines, inadequate staffing levels, imbalanced skill mix, and overall workplace stress and frustration. While these findings align with existing literature, it is noteworthy—and concerning—that despite decades of research highlighting the influence of such factors, they remain persistent and detrimental to nursing practice. Work values, job satisfaction, and a supportive environment conducive to high-quality care are critical components of an effective professional practice environment [42]. Pursio et al. [43] further argue that supportive environments rely on strong leadership, adequate staffing and resources, collegial relationships, and clearly defined quality standards embedded within organisational structures. Nevertheless, recent studies [12,44,45] continue to demonstrate that resource constraints—including limited time, inadequate staffing, and poor skill mix—negatively affect nurses' SoP, sometimes rendering the achievement of quality standards unfeasible within a *"context of scarcity"* (p. 10) [44]. Conceptualising nursing as a *"systems event"* [46] highlights that SoP is not a static set of tasks but rather a dynamic outcome of interactions within a complex care system. It is therefore essential to further explore and address the systemic and environmental modulators that increasingly undermine the scope and quality of nursing practice—particularly those detrimental to patient safety and care standards.

Our findings clearly indicate that nurses' professional stance is shaped—and often constrained—by the quality of the surrounding practice environment. RNs frequently described having to simplify their SoP, despite being aware of the expectations tied to their roles. This dissonance often led to feelings of guilt and inadequacy, reflecting experiences of

moral distress—when individuals are unable to act according to their ethical beliefs due to external constraints, thus compromising their moral integrity [47,48]. The practice environment has increasingly been recognised as a key factor contributing to missed nursing care [49], and as Tan et al. [50] note, moral distress often arises in settings that prevent nurses from practising professionally. Our findings suggest that such challenges may negatively affect nurses' professional identity—comprising core ethical values, knowledge, leadership, and behaviour [51]. Supporting nurses to strengthen their professional identity and self-esteem may help reduce burnout and psychological distress, both of which remain widespread among nurses globally [52]. This is especially important in environments where role expectations are unclear or where nurses experience professional isolation, factors known to undermine the development and sustainability of a strong professional identity [53].

Consistent with existing research, our findings reaffirm the clear distinction between RNs and non-RNs in their involvement with direct and indirect care [54], shaped by their respective SoP in resident care. RNs frequently described their limited presence in the units and in direct care due to their role and function. Their absence from bedside care was largely attributed to time constraints, which they found frustrating. In contrast, non-RNs reported being constantly present and engaged in direct resident care. This contrast reflects the ongoing shift positioning RNs further from the bedside, with a focus on indirect care tasks like medication management, care planning, and documentation. Previous research has noted that RNs often spend a substantial portion of their time on non-value-added tasks—such as searching for supplies, restocking consumables, or locating colleagues— instead of applying their expertise in direct care provision [55]. This results in suboptimal utilisation of RNs' full SoP [56], while simultaneously increasing their administrative and medical responsibilities [57]. These findings underscore the need for clearer delineation and streamlining of the RN SoP to prioritise clinical, supervisory, and care-coordinating functions, while reducing non-essential, non-clinical workload.

A notable divergence in care priorities emerged regarding the FoC. RNs reported that their SoP mainly involved somatic and technical tasks—such as treatments and delegated medical care—despite acknowledging the importance of holistic care, including psychosocial and relational needs. This prioritisation appeared not as a matter of personal choice but as a consequence of systemic constraints. In contrast, non-RNs articulated a broader, more person-centered approach to the FoC, describing their involvement in addressing residents' physical, psychosocial, and relational needs. Importantly, our findings challenge narratives from earlier research suggesting that RNs may not view fundamental care as part of their professional remit [9], or that they consider themselves "too clever" or "too posh to wash" [58,59]. Rather, the RNs in this study were "too busy"—overburdened by administrative tasks and indirect care responsibilities—limiting their ability to deliver comprehensive fundamental care. Their awareness of this imbalance, and the discomfort they expressed, reflected a strong sense of professional responsibility. These findings should be interpreted within the context of broader systemic changes in healthcare, rather than attributed to individual attitudes. Understaffing and task overload go beyond logistical challenges and appears to shape RNs' professional experiences and performance while undermining care quality and core nursing values.

Additionally, the marked divergence in the focus of RNs and non-RNs in the delivery of FoC reflects an adaptive response to the increasing complexity and care needs of residents in nursing homes [60]. However, this adaptation appears to be sub optimally managed, leading to further ambiguity in the RN's role and displacement of their SoP in relation to the FoC [57]. As Kitson et al. [61] argue, it remains essential to reduce non-value tasks and begin to advocate for FoC as a core professional responsibility. These findings carry important policy implications for workforce planning, role clarity, and care quality in facility-based care. The systemic constraints that limit RNs' ability to deliver holistic fundamental care—such as understaffing, administrative burden, and poorly defined scopes of practice—require urgent attention at the organizational and policy levels. Rather than attributing gaps in FoC to individual shortcomings, there is a clear need for policies that recognise and support the RN's role in delivering optimal care. This includes revisiting staffing models to ensure appropriate skill mix, investing in leadership development that prioritises relational and psychosocial dimensions of care, and embedding FoC as a valued, measurable, and supported aspect of professional nursing practice. Without

structural reform, the displacement of RNs from essential care activities may continue to erode both care quality and the integrity of the nursing profession in facility-based care settings.

Our data revealed that many nurses struggled to clearly articulate their roles—particularly those tied to the RN title. Comments such as "it's too difficult to explain" or "the role is unclear" reflected both external confusion and internal uncertainty about professional identity [62]. Non-RNs often recognised RNs' heavy workloads and limited visibility but lacked a clear understanding of their professional SoP [63]. This ambiguity is unsurprising, given the long-standing role expansion, blurred professional boundaries, and growing administrative demands placed on RNs [57,64,65]. These factors make it increasingly difficult for RNs to define their own SoP, let alone establish a shared understanding with colleagues. This points to a deeper fragility in how professional identity is internalised. Since professional identity underpins ethics, knowledge, leadership, and role clarity [51], reinforcing it is likely essential for effective teamwork, care continuity, and long-term retention [65]. Future research should explore how RNs in facility-based care settings experience professional identity, especially amid rising complexity and systemic strain. Understanding the enablers and barriers to identity development could help inform strategies to strengthen professional roles and improve care quality in nursing homes.

Although delegation is a core aspect of the RN's role, our findings reveal challenges in transferring accountability. RNs reported significant reliance on non-RNs to carry out essential care tasks, often under suboptimal conditions. The theme "challenges related to the other" highlighted tensions in this relationship, where RNs had to delegate substantial responsibilities while remaining ultimately accountable. Describing non-RNs as the RN's "operational arm" [22] reflects how delegation and supervision have become integral to the RN's SoP [23]. While effective delegation—assigning tasks to capable individuals—is vital for care quality [24], it was often hindered by inconsistent task uptake, concerns over execution, and asymmetries in availability and visibility between nurses. This aligns with previous research describing delegation as a "double-edged sword" [23], contributing to role strain when RNs are accountable for tasks they do not directly perform. Reluctance to delegate, especially under fears of negative outcomes without direct control, is well documented. Scholars have called for greater leadership development to support safe, effective delegation and clarify expectations [24,65]. Our findings support this need: strengthening delegation practices could improve RN–non-RN collaboration and reinforce delegation as a constructive component of the RN's role. Notably, this theme did not emerge in non-RNs' accounts, suggesting they may view delegation differently, focusing on task execution rather than decision-making. This may reflect their position within hierarchical structures, where they have limited responsibility for delegation or being held accountable for their outcomes. Subsequently, non-RNs may, due to their role positioning (direct executor rather than decision-maker) or structural power differences, perceive less systemic risks related to delegation responsibilities.

## Methodological considerations

This study followed several methodological strategies to ensure trustworthiness, transferability, dependability, and confirmability [66]. Our recruitment process targeted informants with varied experiences and characteristics across four distinct nursing home settings, leading to a diverse and contextually rich dataset, as intended in a qualitative descriptive design [28]. Although our sample can be considered relatively homogeneous in terms of years of work experience, it is worth noting that three of the RNs and three of the non-RNs had completed additional shorter educational courses. One of the eight RNs had undertaken specialist education, i.e., MSc. However, our analysis did not indicate that these informants differed from the others in terms of the codes or themes identified. The use of focused individual interviews [29] enabled informants to express their views in a more open and flexible manner compared to structured formats. Our decision to collect non-dyadic data – individual focused interviews [29] – additionally reduced the influence of hierarchical power dynamics between the two professions, thereby allowing them to speak more freely and independently about their experiences and perspectives. We analysed the material using the framework method for non-dyadic qualitative data [30,67], which, in opposition to dyadic qualitative data where the interviews are conducted together, allowed us to conduct systematic comparison on both an individual level as well as within and across dyads. Notably, five of the six identified themes reflected distinct yet complementary

perspectives from each professional group, emerging consistently across all dyads and sites. This thematic recurrence supports the trustworthiness and theoretical sufficiency of our analysis [68]. With a sample size of 16 interviews (8 dyads), our study exceeds the commonly cited threshold, around twelve interviews at which saturation is likely achieved [69]. Indeed, no new codes or themes were identified in our data matrices following the analysis of the fifth dyad (i.e., 10 interviews). However, the transferability of our findings should be considered within the specific context of Norwegian nursing homes operating under the Nordic welfare model. Caution is therefore warranted when applying these results to healthcare systems outside the Nordic countries and maybe especially systems operating with limited resources. Nonetheless, it is important to note that securing large sample sizes is not a primary objective in qualitative research. Instead, the focus is on ensuring that informants offer a diversity of experiences relevant to the phenomenon under investigation [70].

## Conclusions and implications

To our knowledge, this is the first study to explore the SoP from both RNs and non-RNs using a dyadic methodological approach. Previous research typically addresses these roles in isolation. Thus, this study contributes to the literature by offering a dyadic perspective on SoP —an aspect largely overlooked in prior research. Consequently, our findings emphasise the importance of understanding nursing as a practice embedded within a complex system, where modulators such as the context of care strongly influence both individual and shared SoP. Our study reinforces the need to clarify and support professional identity in both RNs and non-RNs. The observed dependency of RNs on non-RNs for executing key care tasks—particularly in FoC—highlights how systemic pressures and role redefinitions are reshaping the nursing profession in facility-based care settings. The RNs' shift toward technical, medico-delegated, and administrative duties leaves less time for direct engagement with the full spectrum of fundamental care. However, contrary to earlier assumptions, our results suggest this shift is not due to attitudinal resistance but rather systemic overload—RNs are not "too posh to wash" but often simply "too busy" [59]. Delegation of care to non-RNs demands more than task transfer—it requires leadership, communication, and mutual understanding of roles and boundaries. Yet, our findings suggest that professional identity, particularly among RNs, remains insufficiently internalised, leading to ambiguity, diminished visibility, and weakened articulation of scope and responsibilities. These factors can impair collaboration, continuity, and care quality. As such, we propose that nurses professional identity should become a focal point of future research and workforce development, particularly in facility-based care. Enhancing professional identity may support nurses in articulating their roles, engaging confidently in delegation, and reclaiming a more balanced, value-driven SoP. Ultimately, reinforcing identity, role clarity, and delegation skills among nurses is not only beneficial for staff wellbeing but essential for sustainable, high-quality care delivery in nursing homes. Our findings additionally underscore the need for organisational and clinical strategies that clarify role boundaries, reinforce the delivery of fundamental care, strengthen professional identity, and support effective and safe delegation practices.

## Supporting information

**S1 Table. SRQR.** Standards for reporting qualitative research.
(DOCX)

**S2 Table. Overview general themes and underpinning quotes.**
(DOCX)

## Acknowledgments

We wish like to express our gratitude to all involved informants and managers at the four nursing homes for their time, openness, and contribution to this study. This study would not emerge without their engagement. This study was shaped by the valuable insights of the professional, public and patient involvement group in the joint research platform CARE between Lovisenberg Diaconal University College, Norway and Karlstads University, Sweden.

This research report constitutes the third strand of the PhD project *FLORENCE* (FundamentaLs Of nuRsing carE professional praCticE). The project is part of a Nordic collaboration between Lovisenberg Diaconal University College (Norway) and Karlstad University (Sweden) and is embedded within the programmatic research platform *Continuity for Quality of Care in Nursing* (CARE). FLORENCE is conducted with multi-methods focusing on nurses' SoP, the FoC, older people and the context of home- and facility-based care. FLORENCE is performed between 2021 and 2026. Prior to submission a GDPR compliant variant of ChatGPT were used to assist with language refinement and sentence structure optimisation in the final draft as well as after the revision of the manuscript. The authors, on both occasions, critically evaluated and revised all content to ensure that the contributions of AI were limited to minor language improvements without affecting the substance of the work.

## Author contributions

**Conceptualization:** Ole Martin Nordaunet, Gunilla Borglin.

**Formal analysis:** Ole Martin Nordaunet, Hanne Aagaard, Cecilia Olsson, Edith Roth Gjevjon, Gunilla Borglin.

**Investigation:** Ole Martin Nordaunet.

**Methodology:** Ole Martin Nordaunet, Hanne Aagaard, Cecilia Olsson, Edith Roth Gjevjon, Gunilla Borglin.

**Project administration:** Ole Martin Nordaunet.

**Supervision:** Hanne Aagaard, Cecilia Olsson, Edith Roth Gjevjon, Gunilla Borglin.

**Visualization:** Ole Martin Nordaunet.

**Writing – original draft:** Ole Martin Nordaunet, Gunilla Borglin.

**Writing – review & editing:** Ole Martin Nordaunet, Hanne Aagaard, Cecilia Olsson, Edith Roth Gjevjon, Gunilla Borglin.

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
