## [Decision Letter · Decision Letter 0]

28 Jul 2025

PONE-D-25-27718Exploring Nurse Dyads’ Experiences of Scope of Practice in Nursing Homes: A Qualitative Descriptive Study from the FLORENCE ProjectPLOS ONE

Dear Dr. Nordaunet,

Thank you for submitting your manuscript to PLOS ONE. After careful consideration, we feel that it has merit but does not fully meet PLOS ONE’s publication criteria as it currently stands. Therefore, we invite you to submit a revised version of the manuscript that addresses the points raised during the review process

We look forward to receiving your revised manuscript.

Kind regards,

Ahmed Abdelwahab Ibrahim El-Sayed

Academic Editor

PLOS ONE

Journal Requirements: 

Additional Editor Comments:

Dear Authors,

Thank you for your original contribution. I have completed my review of your study. The reviewers have identified a few minor issues that need to be addressed before we can proceed toward a positive editorial decision.

Reviewers' comments:

Reviewer's Responses to Questions

**Comments to the Author**

1. Is the manuscript technically sound, and do the data support the conclusions?

Reviewer #1: Yes

Reviewer #2: Yes

Reviewer #3: No

Reviewer #4: Yes

2. Has the statistical analysis been performed appropriately and rigorously? 

Reviewer #1: Yes

Reviewer #2: Yes

Reviewer #3: Yes

Reviewer #4: Yes

3. Have the authors made all data underlying the findings in their manuscript fully available?

Reviewer #1: Yes

Reviewer #2: Yes

Reviewer #3: Yes

Reviewer #4: Yes

4. Is the manuscript presented in an intelligible fashion and written in standard English?

Reviewer #1: Yes

Reviewer #2: Yes

Reviewer #3: Yes

Reviewer #4: Yes

5. Review Comments to the Author

Reviewer #1: Thank you for the opportunity to review your manuscript titled “Exploring Nurse Dyads’ Experiences of Scope of Practice in Nursing Homes: A Qualitative Descriptive Study from the FLORENCE Project.”

Overall, the manuscript is well-written, methodologically sound, and addresses a timely and relevant issue in the context of nursing home care. The focus on nurse dyads and scope of practice adds a unique and important contribution to the field of gerontological nursing and healthcare workforce studies.

Strengths:

The study design (qualitative descriptive) is appropriate and clearly justified for exploring the stated aims.

The sampling and recruitment process were described adequately, with ethical considerations well addressed.

Thematic analysis is rigorous and well articulated, with clear presentation of themes supported by relevant participant quotes.

The study findings are insightful and provide a foundation for improving nurse collaboration and role clarity in long-term care settings.

The manuscript adheres to reporting guidelines, including ethical declarations, COREQ checklist adherence, and data availability statements.

Suggestions for Improvement:

While the study identifies the impact of organizational context and interpersonal dynamics on scope of practice, the discussion could benefit from deeper integration of existing frameworks or models on interprofessional collaboration or team-based care.

It is recommended to consider limitations regarding potential response bias or power imbalances in nurse dyads (e.g., hierarchical relationships between RNs and ENs), which may influence the openness during interviews.

Although the study mentions theoretical saturation, it would strengthen methodological rigor to describe how saturation was operationalized (e.g., through coding journals or data matrices).

The use of AI language tools for editing is transparently disclosed, which is appreciated. Ensure that future use remains compliant with journal policies.

Conclusion:

The manuscript meets PLOS ONE’s criteria for originality, scientific validity, and ethical compliance. With minor revisions, it can offer a meaningful contribution to the field of nursing workforce research.

Reviewer #2: This research explores how registered nurses (RNs) and non-registered nurses (NRNs) working in Norwegian nursing homes perceive and navigate their respective professional roles. Through dyadic interviews, the study uncovers how both groups adapt their practices in response to workplace pressures, occasionally at the expense of maintaining professional standards. RNs tend to engage more in managerial and indirect care tasks, whereas NRNs are more involved in hands-on, holistic care that addresses residents’ physical, emotional, and relational needs. A major theme is delegation, which often leads to confusion about duties and fosters issues of trust. The blurred boundaries between roles contribute to a weakened sense of professional identity, particularly among RNs who find it difficult to define their specific value. Guided by the Fundamentals of Care framework, the study exposes significant discrepancies in how the scope of practice is understood and executed across roles. While the qualitative findings are insightful, their applicability is somewhat restricted by the Nordic context and the homogeneity of the sample. The study would also benefit from a clearer set of practical recommendations. Ultimately, it emphasizes the importance of defining role boundaries, enhancing delegation protocols, and reinforcing professional identity to improve quality of care in nursing home settings.

Reviewer #3: The manuscript is qualitative in nature. The survey was conducted among RNs and NRNs. There were eight dyads of registered nurses (RNs) and non-registered nurses (non-RNs) (n = 16) working in four nursing homes in south-east Norway. Data were collected by means of a structured interview, then coded, and a visualisation was created showing the specific areas of intervention implemented by RNs and NRNs.

The abstract contains all the elements required by the APA standard, also keywords reflecting the content of the publication

The introduction shows the importance of the topic in providing care in nursing homes, the importance of skill-mix and introduces the context of differentiating competencies, according to the current Scope of Practice. The purpose is well defined. There are occasional single spaces between hyphens and words that should not be there.

The methodology sufficiently describes the design of the study, the tools used and the collection of data, their coding, also the inclusion and exclusion criteria.

The results are grouped in clear tables and mapping to areas of standardised coding is also shown. Themes identified in the standard are described in relation to the survey results. The text was supplemented with statements from respondents.

In the manuscript, reference is made to bioethics committee approval

The discussion refers to the study variables and to the literature - giving a broader context for understanding the results.

Dyads varied in the case of RN competencies, in one there is an APN, some nurses also had additional courses, which could affect the workload, shallowing competencies, or taking higher competencies from NRNs, it would be worth highlighting in the limitations. The sample was not large, however, it gives a picture of the situation and the direction to follow to ensure patient needs and safety

Reviewer #4: I have completed my review of your manuscript, "Exploring Nurse Dyads' Experiences of Scope of Practice in Nursing Homes: A Qualitative Descriptive Study from the FLORENCE Project," and have recommended it for acceptance. I wanted to offer you my congratulations on this outstanding and important work.

Your study is a significant contribution to the nursing literature, providing a much-needed, nuanced perspective on the scope of practice in long-term care. The dyadic approach is particularly innovative and yields rich, insightful findings about the differing professional realities of RNs and non-RNs. The rigor of your methodology and the depth of your analysis are commendable.

In the spirit of collegial feedback, I have a few minor suggestions for your consideration that may further enhance the clarity of your manuscript for its final publication:

Clarifying the "Non-RN" Category: To assist your international readers, you might consider adding a sentence in the Methods section to briefly define the typical qualifications for the "non-registered nurse" roles in the Norwegian context. This would provide valuable context for understanding the team composition.

Highlighting the Dyadic Analysis Rationale: Your use of dyadic analysis is a key strength of the study. To make the rationale even more explicit for readers, you could consider adding a sentence explaining why the dyad was the chosen unit of analysis (e.g., to enable a direct and rich comparison of perceptions from colleagues practicing together).

Reflecting on the Asymmetry of Theme 6: Your finding that Theme 6 ("Challenges Related to the 'Others'") was unique to RNs is fascinating. A point for consideration could be to briefly reflect in your discussion on why this theme might have been absent from the non-RNs' narratives. A sentence or two speculating on their perspective would add another layer to your insightful analysis.

It is a very strong manuscript that will undoubtedly be of great interest to the nursing community. I look forward to seeing it published.

6. PLOS authors have the option to publish the peer review history of their article (what does this mean? ). If published, this will include your full peer review and any attached files.

**Do you want your identity to be public for this peer review?** For information about this choice, including consent withdrawal, please see our Privacy Policy .

Reviewer #1: **Yes: ** Anih Kurnia

Reviewer #2: **Yes: ** DR. IBRAHIM IBN SAANA,Ph.D.

Reviewer #3: No

Reviewer #4: **Yes: ** Jalal Khan

---

## [Author Response · Author response to Decision Letter 1]

6 Aug 2025

Dear reviewers,

Thank you for your valuable and insightful comments. We have revised accordingly and feel that the paper is in a better position now as result of the thorough peer-review. Our point-by-point response is attached with the clean and tracked change manuscript. Sincerely, Ole Martin Nordaunet on behalf of the authors.

---

## [Decision Letter · Decision Letter 1]

19 Aug 2025

PONE-D-25-27718R1Exploring Nurse Dyads’ Experiences of Scope of Practice in Nursing Homes: A Qualitative Descriptive Study from the FLORENCE ProjectPLOS ONE

Dear Dr. Nordaunet,

Thank you for submitting your manuscript to PLOS ONE. After careful consideration, we feel that it has merit but does not fully meet PLOS ONE’s publication criteria as it currently stands. Therefore, we invite you to submit a revised version of the manuscript that addresses the points raised during the review process.

We look forward to receiving your revised manuscript.

Kind regards,

Ahmed Abdelwahab Ibrahim El-Sayed

Academic Editor

PLOS ONE

Journal Requirements:

Additional Editor Comments 

Dear Authors,

Thank you for your revised manuscript. Your submission shows substantial improvement; however, in this round of review, the reviewers have raised some issues that need to be addressed before we can further consider your study.

Reviewers' comments:

Reviewer's Responses to Questions

**Comments to the Author**

1. If the authors have adequately addressed your comments raised in a previous round of review and you feel that this manuscript is now acceptable for publication, you may indicate that here to bypass the “Comments to the Author” section, enter your conflict of interest statement in the “Confidential to Editor” section, and submit your "Accept" recommendation.

Reviewer #1: All comments have been addressed

Reviewer #5: All comments have been addressed

2. Is the manuscript technically sound, and do the data support the conclusions?

Reviewer #1: Yes

Reviewer #5: Partly

3. Has the statistical analysis been performed appropriately and rigorously? 

Reviewer #1: Yes

Reviewer #5: Yes

4. Have the authors made all data underlying the findings in their manuscript fully available?

Reviewer #1: Yes

Reviewer #5: No

5. Is the manuscript presented in an intelligible fashion and written in standard English?

Reviewer #1: Yes

Reviewer #5: Yes

6. Review Comments to the Author

Reviewer #1: The authors have adequately addressed all comments raised in the previous review. The Methods section has been substantially clarified, including participant recruitment, data collection, and the detailed iterative analysis process, which strengthens the rigor and transparency of the study.

Overall, the manuscript is now clearly written, methodologically sound, and aligned with the Standards for Reporting Qualitative Research (SRQR). The study provides a valuable contribution to understanding nursing practice in nursing homes. I am satisfied with the revisions and recommend acceptance of this manuscript in PLOS ONE.

Reviewer #5: Compliance with the data sharing policy is a core requirement of PLOS ONE. The author refused to disclose the data on the grounds of "confidential data" and only provided the contact information of the regulatory agency (SIKT/ Swedish Ethics Review Agency). Specific conditions for data access must be provided (such as signing a confidentiality agreement, anonymization processing procedures). Supplement the legal basis for ethical exemption from disclosing data (such as the provisions of the Personal Data Act of Norway).

7. PLOS authors have the option to publish the peer review history of their article (what does this mean? ). If published, this will include your full peer review and any attached files.

**Do you want your identity to be public for this peer review?** For information about this choice, including consent withdrawal, please see our Privacy Policy .

Reviewer #1: No

Reviewer #5: No

---

## [Author Response · Author response to Decision Letter 2]

21 Aug 2025

Dear Editor-in-Chief, Editorial Team and esteemed reviewers at PLOS One,

We wish to extend our gratitude for the reviewers' valuable feedback and support in the revision of our article. We hope that our amendments will merit renewed consideration for publication in PLOS ONE. The amendments have been highlighted in yellow, and we have also included a detailed point-by-point response letter addressing the considerations raised by Reviewer #5.

Sincerely, Ole Martin Nordaunet on behalf of the authors.

---

## [Decision Letter · Decision Letter 2]

18 Sep 2025

PONE-D-25-27718R2Exploring Nurse Dyads’ Experiences of Scope of Practice in Nursing Homes: A Qualitative Descriptive Study from the FLORENCE ProjectPLOS ONE

Dear Dr. Nordaunet,

Thank you for submitting your manuscript to PLOS ONE. After careful consideration, we feel that it has merit but does not fully meet PLOS ONE’s publication criteria as it currently stands. Therefore, we invite you to submit a revised version of the manuscript that addresses the points raised during the review process.

We look forward to receiving your revised manuscript.

Kind regards,

Ahmed Abdelwahab Ibrahim El-Sayed

Academic Editor

PLOS ONE

Journal Requirements:

Additional Editor Comments:

Thank you for your revision. The reviewer panel, in this version, has raised some minor but important concerns that need to be addressed before we can consider your paper for final publication at PLOS ONE

Reviewers' comments:

Reviewer's Responses to Questions

**Comments to the Author**

1. If the authors have adequately addressed your comments raised in a previous round of review and you feel that this manuscript is now acceptable for publication, you may indicate that here to bypass the “Comments to the Author” section, enter your conflict of interest statement in the “Confidential to Editor” section, and submit your "Accept" recommendation.

Reviewer #5: All comments have been addressed

Reviewer #6: (No Response)

Reviewer #7: (No Response)

Reviewer #8: All comments have been addressed

2. Is the manuscript technically sound, and do the data support the conclusions?

Reviewer #5: Yes

Reviewer #6: Yes

Reviewer #7: Yes

Reviewer #8: Yes

3. Has the statistical analysis been performed appropriately and rigorously? 

Reviewer #5: Yes

Reviewer #6: N/A

Reviewer #7: N/A

Reviewer #8: Yes

4. Have the authors made all data underlying the findings in their manuscript fully available?

Reviewer #5: No

Reviewer #6: Yes

Reviewer #7: Yes

Reviewer #8: Yes

5. Is the manuscript presented in an intelligible fashion and written in standard English?

Reviewer #5: Yes

Reviewer #6: Yes

Reviewer #7: Yes

Reviewer #8: Yes

6. Review Comments to the Author

Reviewer #5: (No Response)

Reviewer #6: The only important point in the article is that in the introduction part there is no mention of the Florence project, which is mentioned in the title.

Reviewer #7: I cannot seem to understand the idea of the dyad when actually what matters is the perception of the nurses of their roles and a little bit of in relation to the non-RNs. If you want to continue with the concept of the dyad, then the voice of the non-RNs should be equally "heard" in this paper.

As it looks now, it is mainly the view of the RNs. I would prefer this as well. If you choose this, you might need to revise the title and everything that pertain to the "dyad"

Reviewer #8: This study offers valuable insights into the lived experiences of nurse dyads working in nursing homes, particularly regarding their scope of practice. The qualitative descriptive approach is appropriate for exploring nuanced interpersonal and professional dynamics, and the focus on dyads adds a unique relational lens to the analysis.

However, some references appear incomplete. Please revise it and modify (Ref. 1, 6, 8, 15, 18, 21, etc). You can add available at (add website) and accessed at (add date).

7. PLOS authors have the option to publish the peer review history of their article (what does this mean? ). If published, this will include your full peer review and any attached files.

**Do you want your identity to be public for this peer review?** For information about this choice, including consent withdrawal, please see our Privacy Policy .

Reviewer #5: No

Reviewer #6: **Yes: ** Alireza Nikbakht Nasrabadi

Reviewer #7: No

Reviewer #8: **Yes: ** Prof Dr. Atyat Mohammed Hassan (Professor of Pediatric Nursing- Faculty of Nursing- Assiut University- Egypt

Associate Professor in Nursing Department- College of Applied Medical Sciences- Prince Sattam Bin Abdulaziz University- Saudia Arabia Kingdom

---

## [Author Response · Author response to Decision Letter 3]

18 Sep 2025

Esteemed reviewers, thank for the continued support for our paper. All reviewers comments has been addressed in the response letter and highlighted in the tracked change manuscript. Sincerely, the authors.

---

## [Editor Report · Decision Letter 3]

23 Sep 2025

Exploring Nurse Dyads’ Experiences of Scope of Practice in Nursing Homes: A Qualitative Descriptive Study from the FLORENCE Project

PONE-D-25-27718R3

Dear Author, 

We’re pleased to inform you that your manuscript has been judged scientifically suitable for publication and will be formally accepted for publication once it meets all outstanding technical requirements.

Kind regards,

Ahmed Abdelwahab Ibrahim El-Sayed

Academic Editor

PLOS ONE

Additional Editor Comments:

Dear Authors,

Thank you for the considerable effort you have invested in revising your manuscript. The improvements are evident, and the reviewers’ concerns have been addressed satisfactorily. I am pleased to inform you that your manuscript has been accepted for publication in PLOS ONE. This decision reflects both the scientific merit of your work and the diligence you demonstrated throughout the revision process. Congratulations on this achievement, and I look forward to seeing your article published.

---

## [Editor Report · Acceptance letter]

PONE-D-25-27718R3

PLOS ONE

Dear Dr. Nordaunet,

I'm pleased to inform you that your manuscript has been deemed suitable for publication in PLOS ONE. Congratulations! Your manuscript is now being handed over to our production team.

Kind regards,

on behalf of

Dr. Ahmed Abdelwahab Ibrahim El-Sayed

Academic Editor

PLOS ONE